# Neural Optimizer Equation, Decay Function, and Learning Rate Schedule Joint Evolution

## Abstract

A major contributor to the quality of a deep learning model is the selection of the optimizer. We propose a new dual-joint search space in the realm of neural optimizer search (NOS), along with an integrity check, to automate the process of finding deep learning optimizers. Our dual-joint search space simultaneously allows for the optimization of not only the update equation, but also internal decay functions and learning rate schedules for optimizers. We search the space using our proposed mutation-only, particle-based genetic algorithm able to be massively parallelized for our domain-specific problem. We evaluate our candidate optimizers on the CIFAR-10 dataset using a small ConvNet. To assess generalization, the final optimizers were then transferred to large-scale image classification on CIFAR-100 and TinyImageNet, while also being fine-tuned on Flowers102, Cars196, and Caltech101 using EfficientNetV2Small. We found multiple optimizers, learning rate schedules, and Adam variants that outperformed Adam, as well as other standard deep learning optimizers, across the image classification tasks.

## 1 Introduction

Deep learning optimizers are built for solving optimization problems, where the goal is to find a set of parameters that optimizes a loss function in an efficient amount of time. The optimization landscapes of deep neural networks are vast and complex terrains with steep cliffs, saddle points, plateaus, and valleys (Goodfellow et al., 2016). Being able to efficiently and intelligently maneuver across these landscapes is vital in order to achieve better performance and evaluation. The simplest way to update the weights of a network is through batch Stochastic Gradient Descent (SGD). With the goal of expediting convergence, adaptive methods have been created to efficiently scale the learning rate per parameter, such as RMSProp, AdaGrad (Duchi et al., 2011), and Adam (Kingma & Ba, 2017). Since the success of these adaptive methods, many researchers have explored other possible optimizers that could increase convergence and performance (Dozat, 2016; Zaheer et al., 2018; Ma & Yarats, 2018; Lucas et al., 2018; Chen et al., 2022a). Coupled with the optimizer is the learning rate schedule, which also directly influences the quality of training (Xu et al., 2019; Smith & Topin, 2019; Gotmare et al., 2018).

Since the success of AutoML (He et al., 2021) and neural architecture search (NAS) (Elsken et al., 2019), automated methods have been applied to finding complete deep learning optimizers in the hopes of finding an optimizer that could outperform SGD or Adam as a drop-in replacement (Bello et al., 2017; Carvalho et al., 2022; Chen et al., 2022b). Searching for candidate optimizer functions can be referred to as *neural optimizer search* (NOS). However, previous work in NOS has become outdated in the sense that their proposed search spaces were created with few operations and argument types. We utilize current deep learning optimizer research to update these deficiencies.

We perform NOS by evolving optimizers on a small ConvNet evaluated on the CIFAR-10 (Krizhevsky & Hinton, 2009) dataset, transferring the final optimizers to CIFAR-100 and TinyImageNet (Le & Yang, 2015), and also fine-tuning EfficientNetV2Small (EffNetV2Small) (Tan & Le, 2021) using their published ImageNet1K weights on Flowers102 (Nilsback & Zisserman, 2008), Cars196 (Krause et al., 2013), and Caltech101 (Fei-Fei et al., 2004). Our summarized contributions are: (1) we propose a new dual-joint search space for NOS, allowing for greater exploration of possible optimizers, and the simultaneous exploration of internal decay functions and learning rate schedules; (2) we propose a simple integrity check able to eliminate degenerate optimizers from wasting valuable computational

time; (3) we propose a problem-specific, mutation-only genetic algorithm able to be massively parallelized; and (4) we discover and present a set of new deep learning optimizers, learning rate schedules, and Adam variants that are capable of surpassing Adam as drop-in replacements across multiple image recognition tasks.

## 2 RELATED WORK

NOS is a relatively new and sparsely explored subdomain of AutoML and NAS (Bello et al., 2017; Carvalho et al., 2022; Chen et al., 2022b). Chen et al. (2022b) evolved deep learning optimizers for image classification by utilizing a program-based search space. Carvalho et al. (2022) evolved deep learning optimizers using a grammar-based representation with very few operations and arguments. Finally, Bello et al. (2017) learned deep learning optimizers by maximizing the reward of an recurrent neural network (RNN) controller through reinforcement learning. Our work heavily stems from the success of Bello et al. (2017), which utilized a grammar-based representation containing unary and binary operations. Not only did Bello et al. (2017) learn the weight update equation for optimizers, but also allowed the controller to decay operands from a set of four possible decay schedules. We greatly expand on this in our proposed search space. Previous work in NOS utilized search spaces containing very simple and primitive operations. Although Bello et al. (2017) used a much larger search space than others (Carvalho et al., 2022; Chen et al., 2022b), and simultaneously allowed to decay operands, the search space has become outdated with regards to current deep learning optimizer research.

Since the emergence of SGD, Momentum, Nesterov Momentum (Nesterov, 1983), RMSProp, Ada-Grad (Duchi et al., 2011), and Adam (Kingma & Ba, 2017), many other deep learning optimizers have been created (Nesterov, 1983; Dozat, 2016; Zaheer et al., 2018; Ma & Yarats, 2018; Lucas et al., 2018; Chen et al., 2022a). The formulation and innovation of these new deep learning optimizers can be used for the advancement of a newer and more powerful search space for NOS. For specifics, Appendix A lists the most recent optimizers that have influenced our proposed search space the most.

Instead of using a grammar-based representation for our optimizers, we choose to utilize a computational graph representation based off the NASNet search space (Zoph et al., 2018). The NASNet search space contains free-floating nodes in a computational graph for cell-like architectures. Each node receives an input argument connection, performs an operation, and outputs the final value to be used by subsequent nodes. We choose to utilize this type of representation as it limits bloating, a common occurrence in tree-based grammars (Luke & Panait, 2006), and allows for easy redundancy within the graph as each node can reuse previous nodes, whereas tree-based grammars cannot.

Although reinforcement learning applied to RNN controllers has been successfully used in previous AutoML deep learning problems (Bello et al., 2017; Zoph et al., 2018; Zoph & Le, 2016), we choose to use its antithesis of evolutionary algorithms to search the search space, which also has had great success in similar scenarios (Real et al., 2019; Chen et al., 2022b; Liu et al., 2020; 2021a).

## 3 METHODOLOGY

### 3.1 SEARCH SPACE

Our proposed search space is composed of two integral parts, one for the weight update equation and another for decay functions (which includes learning rate schedules). We also expand upon Bello et al. (2017) by allowing for the inclusion of momentum-type weight updates. Each optimizer took the form of either no momentum, momentum, or Nesterov momentum, listed in Table 1, where $i$ is the $i^{th}$ time-step, $\alpha$ is the learning rate, $U$ is the weight update equation that is searched for by our evolutionary algorithm applied to our proposed search space, $\beta_i$ is the momentum coefficient (cycled between 0.85 and 0.95 during training), and $z_i$ is an intermediate state saved variable.

#### 3.1.1 OPTIMIZER

The weight update equation for each optimizer is represented by a derivative of the NASNet search space containing three components: (1) *operands*, which are analogous to leaf or terminal nodes for tree-based grammars; (2) *hidden state nodes*, which are free-floating nodes able to use any operand or

| NO MOMENTUM | MOMENTUM | NESTEROV |
|---|---|---|
| | $z_{i+1} = \beta_i * z_i - \alpha * U$ | $z_{i+1} = \beta_i * z_i - \alpha * U$ |
| $w_{i+1} = w_i - \alpha * U$ | $w_{i+1} = w_i + z_{i+1}$ | $w_{i+1} = w_i + \beta_i * z_{i+1} - \alpha * U$ |

Table 1: Optimizer update equations of the form no momentum, momentum, or Nesterov momentum.

output from any previous hidden state node; and (3) a designated *output node*, defined to be the final operation before outputting the weight update equation value. Each node either performs a unary or binary operation on its received inputs. The final output value is built by mapping all connections that reach the output node. Because there exists a probability that a particular node does not connect to the output node, it is regarded as *inactive*, while nodes that reach the output node, either through direct connection or by a subsequent node connecting to the output node, are regarded as *active*. Active and inactive nodes allow for the size of the weight update equation to grow and shrink during evolution, not forcing each optimizer to use all the nodes in the graph.

Our 20 proposed operands are listed in Table 2a, where $\hat{v}$, $\hat{s}$, $\hat{\lambda}$ are the running exponential moving averages of $g$, $g^2$, and $g^3$, with $\beta_1 = 0.9$, $\beta_2 = 0.99$, and $\beta_3 = 0.999$. Unlike Bello et al. (2017), we do not bias our optimizers towards Adam or RMSProp by incorporating them as operands. Our 23 proposed scalar unary operations are listed in Table 2b, where drop$(x, p)$ drops the entries of $x$ with probability $p$, and norm$(x)$ scales the tensor $x$ by the L2 norm. Our 10 proposed binary operations are listed in Table 3a, where clip$(x_1, \pm|x_2|)$ clips $x_1$ to the range $-|x_2|$ to $|x_2|$. In addition to simple scalar type unary operators, we also propose unary operators that perform state-holding operations. Specifically, each state-holding unary operator maintains a state variable $z_i$ that is updated at each time-step. The 3 proposed state saving unary operators are shown in Table 3b.

(a) The proposed set of operands.

| OPERANDS | |
|---|---|
| $g$ | $g^2$ |
| $g^3$ | $\hat{v}$ |
| $\hat{s}$ | $\hat{\lambda}$ |
| sign$(g)$ | sign$(\hat{v})$ |
| 1 | 2 |
| $10^{-6}w$ | $10^{-5}w$ |
| $10^{-4}w$ | $10^{-3}w$ |
| $0.3g + 0.7\hat{v}$ | $0.05g^2 + 0.95\hat{s}$ |
| $0.01g^3 + 0.99\hat{\lambda}$ | |
| $\frac{1}{3}\sum_j^3 \beta^j v^j - g$ | for $\beta^j = [0, 0.9, 0.999]$ |
| $\frac{1}{3}\sum_j^3 \beta^j s^j - g^2$ | for $\beta^j = [0, 0.99, 0.999]$ |
| $\frac{1}{3}\sum_j^3 \beta^j \lambda^j - g^3$ | for $\beta^j = [0, 0.999, 0.9999]$ |

(b) The proposed set of scalar unary operations.

| SCALAR UNARY OPERATIONS | |
|---|---|
| $x$ | $-x$ |
| $\ln(|x| + \epsilon)$ | $\sqrt{|x|}$ |
| $e^x$ | $|x|$ |
| sigmoid$(x)$ | $\frac{d}{dx}$sigmoid$(x)$ |
| softsign$(x)$ | $\frac{d}{dx}$softsign$(x)$ |
| softplus$(x)$ | erf$(x)$ |
| $\tanh(x)$ | arctanh$(x)$ |
| bessel$_{i1e}(x)$ | arcsinh$(x)$ |
| max$(x, 0)$ | min$(x, 0)$ |
| drop$(x, 0.5)$ | drop$(x, 0.3)$ |
| drop$(x, 0.1)$ | norm$(x)$ |
| erfc$(x)$ | |

Table 2: The proposed sets of operands and scalar unary operations for the optimizer update equation search space.

(a) The proposed set of binary operations.

| BINARY OPERATIONS | |
|---|---|
| $x_1 + x_2$ | $x_1 * x_2$ |
| $x_1 - x_2$ | $x_1/(x_2 + \epsilon)$ |
| $x_1/\sqrt{1 + x_2^2}$ | max$(x_1, x_2)$ |
| min$(x_1, x_2)$ | $0.95x_1 + 0.05x_2$ |
| clip$(x_1, \pm|x_2|)$ | $|x_1|^{x_2}$ |

(b) The proposed set of state saving unary operations.

| STATE SAVING UNARY OPERATIONS |
|---|
| $z_{i+1} = 0.95x_i + 0.05z_i$ |
| $z_{i+1} = x_i - z_i$ |
| $z_{i+1} = \max(x_i, z_i)$ |

Table 3: The proposed sets of scalar binary operations and state saving unary operations for the optimizer update equation search space.

Each computational graph is initialized with four hidden state nodes and one output node. Each node is initialized by sampling its operation from the set of all binary and unary operations. Each

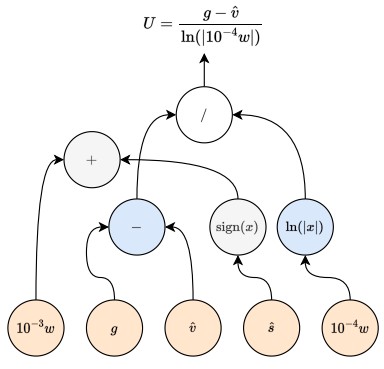 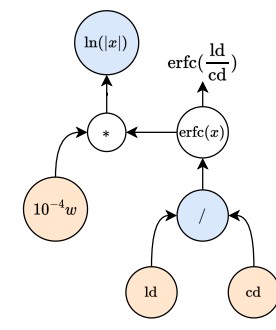

(a) Example Optimizer Update Equation

(b) Example Decay Function

Figure 1: (a) Example optimizer graph with two active (blue) hidden state nodes, two inactive hidden state nodes (grey), and one root node (white). The final weight update equation is given above the root node. Note that this does not include momentum type. (b) Example decay function applied to the $10^{-4}w$ operand before being applied in the $\ln(|x|)$ operation. The decay graph contains one active (blue) hidden state node, zero inactive hidden state nodes (grey), and one root node (white). The final decay function equation is given above the root node.

connection for each hidden state node was sampled from all input connections and hidden state nodes. The momentum type of the weight update was sampled uniformly from the three choices mentioned in Section 3.1. Lastly, each connection within the graph had a probability of generating a decay function. Figure 1a shows an example optimizer's weight update equation after random initialization.

### 3.1.2 DECAY FUNCTION

A decay function is a scheduled function with the ability to decay operands or hidden state nodes within the optimizer graph. Unlike Bello et al. (2017), we do not apply a simple decay function to operands and outgoing connections, but allow for the creation of decay functions through a separate search space. This subsequent search space is applied to each argument connection between hidden state nodes in the weight update graph. Much like our proposed search space for the weight update, the search space for decay functions is represented as a computational graph with free-floating nodes, along with the same properties as before. The operands in this search space refer to decay schedules. The binary operations stay the same; however, we limit the unary operations to operations with an upper limit of 1 for values between 0 and 1. This was performed to prevent scaling input argument values to greater than their original value. The updated choices of unary operators for the decay functions are shown in Appendix B.

Unlike Bello et al. (2017), we do not limit our decay schedules to decay only, but also include their increase as well. Our proposed schedules are the following: linear decay ($ld$), linear increase ($li$), linear decrease with restart ($ldr$), linear increase with restart ($lir$), cosine decay ($cd$), cosine increase ($ci$), cosine decrease with restart ($cdr$), cosine increase with restart ($cir$), cyclic cosine decay ($ccd$), cyclic cosine increase ($cci$), exponential decay ($ed$), exponential increase ($ei$), demon decay ($dd$) (Chen et al., 2022a), and demon increase ($di$). In total, there are 14 different possible schedules to be used as operands within the decay search space graph, shown in Appendix B.

Lastly, the decay functions can be used to create unique learning rate schedules. Unless the decay function is applied inside a non-distributive function, such as $\tanh(x)$, the decay function can be factored out alongside the learning rate. As a result, the proposed complete search space can optimize not only the weight update equation, but also the learning rate schedule and internal decay functions for non-distributive functions.

Each computational graph for the decay functions is initialized with one hidden state node and one output node. This was performed to simplify the decay schedules. Each node in the decay function graph was initialized similar to the weight update equation graph. Figure 1b shows an example decay function being applied to a connection from within the example optimizer graph from Figure 1a.

## 3.2 INTEGRITY CHECK

All together, the proposed search for both the weight update equation and decay functions contain an enormous number of possible combinations, creating an extremely sparse search space for workable optimizers. To prevent degenerate optimizers from wasting valuable evaluation time, an integrity check was implemented from the motivation of Liu et al. (2021b) in their evolution of loss functions. The integrity check needs to be accurate in detecting degeneracy while also being simple to compute. Our proposed integrity check was based on the shifted sphere optimization problem, where the goal is to minimize $f(x) = \sum_i^n (x_i - \beta_i)^2$ for $n$ variables, where $\beta_i$ is a shift constant for the $i^{th}$ variable.

Each optimizer was tested at minimizing the optimization problem with different learning rates. Each optimizer was given the same initial point and was run for a set number of iterations. If none of the final objective function values for each learning rate was below a hand-designed threshold, it was rejected as it was unable to achieve moderate convergence on a simple problem. The chosen initial point, associated shift constants, number of variables, number of iterations, and threshold were hand designed based on the success from current deep learning optimizers.

Upon initialization, if a connection was sampled to contain a decay function, the associated decay function underwent a simpler integrity check: the decay function was rejected if it scaled the component to less than 0 or more than 1. The decay function looped initialization until it found a graph that satisfied this integrity check.

## 3.3 SURROGATE FUNCTION

With the goal of learning optimizers that perform well on large-scale deep learning models, evaluating each candidate optimizer on a large-scale model can be extremely computationally expensive. Surrogate functions are cheap proxy models used to estimate the performance of deep learning components at large-scale (Liu et al., 2020; Bingham et al., 2020; Bello et al., 2017). From the success of Bello et al. (2017), we also choose to utilize a small ConvNet as our surrogate function. The ConvNet contains three convolution layers, followed by batch normalization (Ioffe & Szegedy, 2015) and Swish activation (Ramachandran et al., 2017). Except for the first convolution layer, each subsequent convolution layer maintained a stride of 2, and doubled the previous number of filters/channels. The starting number of filters/channels for the first convolution layer was referred to as the *base*.

## 3.4 EARLY STOPPING

Even though an integrity check was implemented to prevent degenerate optimizers from wasting valuable evaluation time, some optimizers may not perform well for deep learning tasks. To prevent these optimizers from wasting valuable time, two early stopping mechanisms were implemented. First, each optimizer was trained on the ConvNet with *base* 48 (totaling to 0.212M parameters) for 800 steps at each of the following learning rates: 10, 1, 0.1, 0.01, 0.001, 0.0001, and 0.00001. In addition, a one cycle cosine decay learning rate schedule with a linear warmup (Smith, 2017) was applied to every optimizer. As a result, all optimizers were learning to change the given learning rate schedule using their decay functions during evolution. If none of the final training accuracies from each learning rate session yielded an accuracy above a hand-designed threshold of 25%, the optimizer was rejected. Second, if the optimizer passed the first early stopping mechanism, the optimizer was trained again on the ConvNet, except for 8,000 steps using the best found learning rate. After 1,000 steps, if the training accuracy ever fell below a hand-designed threshold of 40%, training was stopped and the best validation accuracy was returned as the fitness value. If the optimizer passed both early stopping criteria, the best validation accuracy was used as the optimizers final fitness value. For the validation dataset, 5,000 of the 50,000 training images for CIFAR-10 were set aside as validation.

## 3.5 PARTICLE-BASED GENETIC ALGORITHM

Real et al. (2019) applied *regularized evolution*, a mutation-only, population-based genetic algorithm (GA), to the NASNet search space with great success, but we found it non-optimal for our problem. In some preliminary runs, *regularized evolution* converged extremely fast. We believe that this occurrence was due to our extremely sparse search space, paired with our integrity check, creating an extremely sharp local minima for each optimizer. As a result, only a few non-degenerate solutions exist nearby each optimizer, which are quickly exhausted within a few iterations of the algorithm.

To address this problem, we propose a problem-specific, particle-based GA with the same components as *regularized evolution*, namely: (1) mutation only, (2) aging, and (3) parallelism. In our algorithm, $n$ independent particles run for $t$ time-steps, where at each time-step $k$ random mutations are performed. At each timestep, each particle is mutated $k$ times (mutation only) and the best mutated child is selected as the next position for the particle (aging). Each particle acts independently so that the available search space around each optimizer can be quickly explored and exhausted. Our proposed GA allows for embarrassing parallelization (parallelism) at two levels. First, because each particle is independent from the other, as there is no competition, each particle can be parallelized. Second, because each particle performs $k$ independent mutations at each timestep, each mutated child can be parallelized. Therefore, the only bottleneck in our proposed GA is $t$, the total number of timesteps, which is relatively small as the available search space around optimizers is quickly exhausted. If $n * k$ GPUs are available, the running time of the algorithm is $O(t * c)$ where $c$ is the mean cost to train a neural network. See Appendix C for pseudocode.

Because our search space is extremely sparse, a poor initial point would be likely to generate poor mutations. As a possible solution, we use an enlarged initial population. For each run, we generate $10n$ randomly initialized optimizers and evaluate them on a ConvNet with *base*=32. The best $n$ were then taken to be the initial points for each run. We ran our particle-based GA three times, each with different configurations. With the goal of having each run evaluate approximately 1,800 optimizers over the course of evolution, we ran the following three configurations: (1) $n = 50$, $k = 6$, $t = 6$; (2) $n = 50$, $k = 5$, $t = 7$; and (3) $n = 50$, $k = 7$, $t = 5$. Although our proposed algorithm allows for massive parallelization, it was performed sequentially as we only had access to one NVIDIA 4090 GPU during evolution. Each run took approximately six days to complete.

Mutation was performed by selecting an active node randomly, and then performing one of six possible mutations. First, the operation was mutated by randomly sampling the list of remaining available operations. Second, the argument connection was mutated by randomly sampling the list of remaining available arguments and hidden state nodes. Third, a unary operation was changed to a binary operation, or visa-versa. For unary to binary, the extra connection was sampled randomly from the list of possible connections (either argument connections or other hidden state nodes); for binary to unary, one of the connections was randomly dropped. Fourth, which was only available for binary operations, the argument connections were swapped. Fifth, the momentum type of the weight update was sampled from the list of remaining types. Lastly, the decay schedule for one of the connections of an active node was mutated. If a decay schedule was not present for the given connection at time of mutation, a randomly initialized decay schedule was assigned. If a decay schedule was present, it was either deleted or mutated. If mutated, the computational graph of the decay schedule was mutated using the operations one through four.

Each mutation was passed through the integrity check, from Section 3.2, to ensure non-degeneracy. If the integrity check failed, mutation was looped until a child mutation was found that passed the integrity check. All mutated child optimizers were then evaluated using the surrogate function described in 3.3. If the optimizer failed the first early stopping mechanism, mutation was performed again until a child mutation was found that passed the first early stopping mechanism.

Lastly, after performing our evolutionary experiments, we performed an optimizer elimination protocol that progressively eliminated optimizers based upon their performance on increasingly larger models to ensure the final optimizers could correlate to well to larger models. This procedure is detailed in Appendix D.

## 3.6 ADAM VARIANTS

We performed two supplementary experiments with the goal of (1) obtaining variants of the Adam optimizer and (2) variants of learning rate schedules for Adam. For our first experiment, we hand programmed the Adam optimizer into our search space and ran our proposed GA using Adam as the initial particle. It ran using $k = 16$, $t = 3$, and *base* 48. We allowed all components to be mutated. For our second experiment, we used the Adam optimizer again as the initial particle to our proposed GA, except this time only the decay schedules were allowed to be mutated for each connection, not the weight update equation. It ran using $k = 12$, $t = 4$, and *base* 48.

# 4 RESULTS

## 4.1 FINAL OPTIMIZERS

The final 10 optimizers discovered after optimizer elimination are listed in Table 4. There appear to be three families of optimizers present in the final results, with two outsiders. All optimizers, except the two outsiders (Opt9 and Opt10), used the quasi-hyperbolic momentum (Ma & Yarats, 2018) update equation ($0.3g + 0.7\hat{v}$), showcasing the importance of a linear combination between the gradient and its exponential moving average. The first family of optimizers (Opt1, Opt2, and Opt3) took the form ($0.3g + 0.7\hat{v}$) + softsign($x$), where $x$ was dependent upon the optimizer. This family heavily relied upon a weight decay of $10^{-5}w$ inside the softsign operation. The second family of optimizers (Opt4, Opt5, and Opt6), took the form ($t_i^t(0.3g + 0.7\hat{v}))/(t_i^2 x)$ where $x$ was dependent upon the optimizer. This family heavily relied upon decay functions, with the majority directly affecting the learning rate schedule. The last family of optimizers (Opt7 and Opt8) are exactly the same, except for Opt8 trading the outside tanh($x$) function of Opt7 for arcsinh($x$). Lastly, the two outsiders (Opt9 and Opt10) seem to be descendants from two non-included families of optimizers; however, they are the only two final optimizers to incorporate a momentum type. Seven of the final ten optimizers contain decay functions, with four of them directly effecting the learning rate schedule.

| Name | Momentum | Weight Update Equation | Decay Functions |
|---|---|---|---|
| Opt1 | None | $(0.3g + 0.7\hat{v})+$ softsign(clip($10^{-5}w$, $10^{-5}w - (0.05g^2 + 0.95\hat{s})$))) | None |
| Opt2 | None | $(0.3g + 0.7\hat{v})+$ softsign($\frac{10^{-5}w}{\sqrt{1+(10^{-5}w-(0.05g^2+0.95\hat{s}))^2}}$) | None |
| Opt3 | None | $(0.3g + 0.7\hat{v})+$ softsign($\frac{10^{-5}w}{\sqrt{1+(10^{-5}w-(0.01g^3+0.99\hat{\lambda}))^2}}$) | None |
| Opt4 | None | $\frac{t_i^1(0.3g+0.7\hat{v})}{t_i^2\text{clip}(2,t_i^3 e^{\hat{v}})}$ | $t_i^1 = \text{erfc}(\text{erfc}(ci))$ $t_i^2 = \frac{d}{dx}\tanh(cir)$ $t_i^3 = \arctan(dd)$ |
| Opt5 | None | $\frac{t_i^1(0.3g+0.7\hat{v})}{t_i^2\text{clip}(2,e^{\hat{v}})}$ | $t_i^1 = \text{erfc}(\text{erfc}(ci))$ $t_i^2 = \frac{d}{dx}\tanh(cir)$ |
| Opt6 | None | $\frac{t_i^1(0.3g+0.7\hat{v})}{t_i^2\|t_i^3 e^{10^{-4}w}\|}$ | $t_i^1 = \text{erfc}(\text{erfc}(ci))$ $t_i^2 = \frac{d}{dx}\tanh(cir)$ $t_i^3 = \frac{d}{dx}\tanh(ci)$ |
| Opt7 | None | $\tanh(t_i^1\text{arcsinh}(0.3g + 0.7\hat{v}))$ | $t_i^1 = \max(cci, lir)$ |
| Opt8 | None | $\text{arcsinh}(t_i^1\text{arcsinh}(0.3g + 0.7\hat{v}))$ | $t_i^1 = \max(cci, lir)$ |
| Opt9 | Nesterov | $t_i^1 g e^{\arctan(0.05g^2+0.95\hat{s})}$ | $t_i^1 = \text{erfc}(ed)$ |
| Opt10 | Nesterov | $\text{bessel}_{i1e}(\text{bessel}_{i1e}(t_i^1 g))$ | $t_i^1 = dd * li$ |

Table 4: Final 10 optimizers found after evolution. The momentum type, weight update equation, and decay functions used by each optimizer are listed. We refer to each optimizer in the paper by the associated name.

## 4.2 SUPPLEMENTARY EXPERIMENTS

The final best five optimizers found from our first supplementary experiment for finding variants of Adam are shown in Table 5a. As one can see, our algorithm swapped the division operator between $\hat{v}$ and $\hat{s}$ in favor of the clip operator, as well as replacing the square root operator for $\hat{s}$ with some other function. From our second supplementary experiment on finding learning rate schedules for Adam, we took the best schedules found, along with the best learning rate schedules found during our standard evolution, to report in Table 5b. Note that each learning rate schedule present is multiplied by $lr_{cd}$, the cosine learning rate decay schedule with linear warmup mentioned in Section 3.4. See Appendix F for the plots of the discovered learning rate schedules and internal decay functions.

|  | (a) Adam Variants |  | (b) Learning Rate Schedules |  |
|---|---|---|---|---|

**Adam Variants**

| A1 | clip($\hat{v}, \sqrt{\hat{s}}$) |
|---|---|
| A2 | clip($\hat{v}, |\ln(\hat{s})|$) |
| A3 | clip($\hat{v}, \sqrt{|\ln(\hat{s})|}$) |
| A4 | clip($\hat{v}, \text{sigmoid}(\hat{s})$) |
| A5 | norm(clip($\hat{v}, \sqrt{\hat{s}}$)) |

**Learning Rate Schedules**

| LR1 | $\dfrac{\text{erfc}(\text{erfc}(ci))}{\frac{d}{dx}\tanh(cir)}$ | LR6 | $\dfrac{\arctan(li)\text{erfc}(cci)}{\sqrt{\frac{d}{dx}\text{softsign}(cci)}}$ |
|---|---|---|---|
| LR2 | $\dfrac{\text{erfc}(\text{erfc}(ci))}{\frac{d}{dx}\tanh(cir)*\frac{d}{dx}\tanh(ci)}$ | LR7 | $\dfrac{1}{\sqrt{\frac{d}{dx}\text{softsign}(lir)}}$ |
| LR3 | $\dfrac{\arctan(li)}{\frac{d}{dx}\tanh(lri)*\sqrt{\frac{d}{dx}\text{softsign}(di)}}$ | LR8 | $\dfrac{1}{\sqrt{\text{softsign}(\arctan(ei))}}$ |
| LR4 | $\dfrac{\text{sigmoid}(li)^2}{\text{sigmoid}(2*\text{softsign}(ld))}$ | LR9 | $\tanh(\max(cci, lri))$ |
| LR5 | $\dfrac{1}{\sqrt{\text{erf}(ed)}}$ | | |

Table 5: (a) The best five Adam variants found during our first supplementary experiment. Note that all optimizers listed here did not use a momentum type. (b) The best nine learning rate schedules found from standard evolution and our supplementary experiment on finding learning rate schedules for Adam.

## 5 TRANSFERABILITY EXPERIMENTS

To assess generalization, all found optimizers, Adam variants, and learning rate schedules (trained on Adam) were transferred to various image classification tasks. Each were trained on CIFAR-10 and CIFAR-100 from scratch using EffNetV2Small with progressive RandAug regularization strategy. Each were also trained on TinyImageNet (Tiny) using a custom ResNet9 (6.5M parameters). For further study on fine-tuning scenarios, each were trained for fine-tuning EffNetV2Small using its official ImageNet1K weights on Flowers102, Cars196, and Caltech101. For comparison, the results for Adam, RMSProp, SGD, and Nesterov momentum are recorded as well for each experiment. In addition, as a baseline comparison to Bello et al. (2017), we have included their best two discovered optimizers, PowerSign-ld (linear decay) and AddSign-ld (linear decay). Lastly, because many of our discovered optimizers heavily utilize the quasi-hyperbolic momentum term, we also include the original QHM (Ma & Yarats, 2018) optimizer for comparison. To assess how much the decay functions influence the quality of the found optimizers containing them, each were re-run without them. Opt4 was retrained twice, once without using $t_i^1$ and $t_i^2$ (Opt4$_1$); and a second time without using any $t_i$ (Opt4$_2$). Opt6, Opt7, Opt8, Opt9, and Opt10 were all retrained without using any $t_i$ (Opt6$_1$, Opt7$_1$, Opt8$_1$, Opt9$_1$, Opt10$_1$). See Appendix G for exact implementation and training details for each experiment. The results for all image recognition experiments are recorded in Table 6. In addition to image recognition, a supplementary experiment was performed where the final optimizers were evaluated on language modeling for the PTB (Marcus et al., 1993) dataset. Those results are discussed in Appendix H.

## 6 RESULTS AND DISCUSSION

From Table 6, one can see that many optimizers, learning rate schedules, and Adam variants were able to outperform Adam and the other standard deep learning optimizers across the image recognition datasets. The results can be broken down into two sections: those trained from scratch (CIFAR and Tiny) and those fine-tuned (Flowers, Cars, and Caltech). When training from scratch on CIFAR and Tiny, Opt4, Opt5, Opt6, Opt7, Opt8, and Opt10 performed relatively well compared to the baseline deep learning optimizers. However, Opt6 elevated itself above the rest by always being in the Top 3. Opt6$_1$ is the quasi-hyperbolic momentum update equation ($0.3g + 0.7\hat{v}$) scaled by exponential of the weight decay $\exp(10^{-4}w)$. For small weight values, the exponential weight decay approaches one, making Opt6 equivalent to standard quasi-hyperbolic momentum. However, Opt6 simultaneously learned decay functions. These decay functions can be factored out along side the learning rate, giving rise to LR2. The success of Opt6 is heavily dependent upon the inherently learned LR2 schedule, as Opt6$_1$ always under-performed Opt6 when training from scratch. Opt10$_1$ always out-performed Nesterov's momentum when training from scratch. Appendix F discusses the effect of the scaling on the gradients for Nesterov. We empirically noticed that Opt10$_1$ liked large learning rates around 10. We believe that the double scaling of the gradients clips larger gradient values, allowing for larger learning rates to scale gradients near zero to have more of an effect, which empirically seems beneficial. When fine-tuning using transfer learning, there appears to be a flip in performance between the optimizers, as Opt1, Opt3, A1, and A5 achieved Top 3 atleast once

across Flowers, Cars, and Caltech. Opt1, Opt2, and Opt3 can be seen as extensions of QHM as they are detailed by QHM+$x$, where $x$, was dependent upon the optimizer. Although the results for Opt1-Opt3 are similar to QHM for fine-tuning, the results from CIFAR reveal that the additional terms are beneficiary as Opt1-Opt3 all outperformed QHM. Appendix F discusses the effect of each learned term to the weight update equation. For the Adam variants, the majority of their success occurs when fine-tuning, as all achieved Top 8 once across the three datasets. Not regarding CIFAR-10, A1 and A5 outperformed Adam across all other datasets, giving empirical evidence to the swapping of the division operator with the clip, along with adding a normalization operator for A5.

| OPTIMIZER | CIFAR-10 | CIFAR-100 | FLOWERS | CARS | CALTECH | TINY |
|---|---|---|---|---|---|---|
| ADAM | 95.69±0.15 | 77.24±0.33 | 97.36±0.14 | 91.30±0.16 | 91.41±0.11 | 47.43±0.82 |
| RMSPROP | 95.55±0.05 | 77.62±0.23 | 97.44±0.21 | 90.68±0.16 | 91.53±0.34 | 46.93±0.38 |
| SGD | 95.04±0.19 | 77.34±0.30 | 97.48±0.07 | **91.39±0.21** | **92.76±0.44** | 44.60±0.27 |
| NESTEROV | 95.21±0.22 | 77.78±0.24 | **97.73±0.18** | 91.29±0.17 | 91.52±0.28 | **48.38±0.44** |
| POWERSIGN | 95.51±0.25 | 78.01±0.26 | **97.61±0.18** | 90.39±0.24 | **92.23±0.19** | 47.84±0.27 |
| ADDSIGN | 95.27±0.14 | 78.15±0.15 | 97.52±0.17 | 90.52±0.16 | 91.87±0.25 | **48.15±0.18** |
| QHM | 95.00±0.10 | 77.45±0.18 | **97.66±0.19** | **91.31±0.16** | 92.10±0.25 | 47.49±0.20 |
| OPT1 | 95.58±0.19 | 78.55±0.18 | 97.45±0.09 | **91.47±0.28** | **92.22±0.10** | 47.46±0.41 |
| OPT2 | 95.48±0.20 | 78.87±0.31 | 97.36±0.11 | **91.54±0.09** | 92.16±0.29 | 47.56±0.40 |
| OPT3 | 95.47±0.27 | 78.95±0.29 | **97.76±0.12** | **91.54±0.20** | 92.15±0.29 | 47.83±0.73 |
| OPT4 | **95.95±0.06** | **79.31±0.15** | 97.24±0.12 | 89.36±0.36 | **92.23±0.25** | 47.96±0.81 |
| - OPT4$_1$ | 95.21±0.17 | 77.45±0.27 | 97.35±0.09 | **91.40±0.21** | 92.06±0.10 | 45.88±0.56 |
| - OPT4$_2$ | 95.48±0.12 | 77.42±0.37 | 97.45±0.09 | 90.33±0.22 | 92.10±0.20 | 45.13±0.92 |
| OPT5 | **96.02±0.19** | **79.05±0.37** | 97.23±0.21 | 89.08±0.10 | 92.18±0.23 | 47.80±1.02 |
| OPT6 | **96.16±0.16** | **79.46±0.36** | 97.27±0.17 | 89.06±0.23 | 91.71±0.32 | **48.51±0.42** |
| - OPT6$_1$ | 94.98±0.15 | 77.55±0.12 | 97.51±0.15 | 88.87±0.17 | 92.17±0.45 | **48.20±0.53** |
| OPT7 | 95.82±0.07 | **79.44±0.21** | 97.25±0.12 | 89.32±0.19 | 91.65±0.15 | 47.92±0.54 |
| - OPT7$_1$ | 94.99±0.09 | 77.45±0.26 | 97.37±0.11 | 88.94±0.21 | 92.18±0.28 | **48.23±0.50** |
| OPT8 | 95.74±0.09 | **79.23±0.29** | 97.26±0.15 | 89.46±0.17 | 91.98±0.25 | 47.96±0.19 |
| - OPT8$_1$ | 95.02±0.20 | 77.47±0.41 | **97.59±0.09** | 89.07±0.10 | 92.08±0.22 | 47.91±0.36 |
| OPT9 | 95.85±0.19 | **79.06±0.49** | 97.54±0.15 | 90.57±0.21 | 91.08±0.35 | 46.64±1.64 |
| - OPT9$_1$ | 95.64±0.18 | 77.51±0.23 | 97.51±0.13 | 90.92±0.07 | 91.42±0.32 | 47.44±0.66 |
| OPT10 | **96.04±0.13** | **79.80±0.27** | 97.11±0.16 | 89.20±0.19 | 91.97±0.29 | 47.60±0.41 |
| - OPT10$_1$ | 95.51±0.18 | 78.12±0.11 | 97.32±0.13 | 90.58±0.10 | 91.99±0.16 | **48.82±0.48** |
| LR1 | **95.99±0.15** | 77.64±0.20 | 97.31±0.14 | 90.53±0.10 | 91.36±0.49 | 43.23±0.29 |
| LR2 | **96.13±0.10** | 78.10±0.17 | 97.18±0.24 | 90.25±0.30 | 91.21±0.43 | 43.31±0.19 |
| LR3 | **96.23±0.07** | 78.48±0.07 | 97.29±0.23 | 90.56±0.04 | 91.20±0.34 | 41.89±0.27 |
| LR4 | 95.57±0.06 | 77.15±0.15 | 97.53±0.24 | 90.45±0.26 | 90.90±0.24 | 44.80±0.32 |
| LR5 | **95.97±0.13** | 78.55±0.45 | 96.72±0.20 | **91.47±0.09** | 88.99±0.19 | 47.04±0.65 |
| LR6 | 94.57±0.11 | 74.42±0.29 | 97.41±0.16 | 89.63±0.36 | 91.76±0.16 | 39.81±0.76 |
| LR7 | 95.63±0.12 | 77.31±0.40 | 97.23±0.20 | 91.79±0.21 | 89.93±0.14 | 46.84±0.38 |
| LR8 | 95.28±0.15 | 76.64±0.23 | 97.21±0.08 | 91.74±0.11 | 89.47±0.35 | **48.20±0.32** |
| LR9 | 95.76±0.05 | 77.69±0.36 | **97.61±0.20** | 90.61±0.17 | 91.39±0.35 | 43.27±0.45 |
| A1 | 95.26±0.16 | 77.50±0.22 | **97.66±0.13** | **91.36±0.15** | **92.20±0.28** | 47.92±0.45 |
| A2 | 95.17±0.16 | 77.60±0.35 | 97.19±0.18 | 91.22±0.20 | **92.21±0.12** | 48.12±0.40 |
| A3 | 95.04±0.23 | 77.04±0.29 | 97.50±0.09 | 91.29±0.06 | **92.20±0.16** | 47.65±0.22 |
| A4 | 95.04±0.19 | 77.29±0.27 | 97.50±0.18 | 91.30±0.13 | **92.21±0.13** | 48.15±0.43 |
| A5 | 94.95±0.38 | 77.55±0.23 | **97.65±0.21** | **91.60±0.14** | 91.92±0.18 | **48.40±0.50** |

Table 6: Results for all optimizers and learning rate schedules for CIFAR-10, CIFAR-100, Flowers102 (Flowers), Cars196 (Cars), Caltech101 (Caltech), and TinyImageNet (Tiny). The mean and standard deviation of the test accuracy across 5 independent runs are reported. Red indicates Top 3 performance, while bold indicates Top 8 performance.

## 7 CONCLUSIONS

In this work, we expanded on previous research in NOS by proposing a new search space containing the most up-to-date research from deep learning optimizers. This new search space allows for the simultaneous optimization of the weight update equation, internal decay functions, and adaptation of the learning rate schedule. We searched the space using our proposed particle-based GA that is able to be massively parallelized. We save computational resources by incorporating an integrity check to get rid of degenerate optimizers with our initialization and mutation operator in order to make intelligent mutational choices. In addition, we perform two supplementary experiments to obtain Adam variants and new learning rate schedules for Adam. After transferring the final optimizers, we found five new optimizers and two Adam variants, that consistently outperformed standard deep learning optimizers when training from scratch and fine-tuning on image classification tasks: Opt1, Opt3, Opt4, Opt6, Opt10, A1, and A5.

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

## A EXTENDED RELATED WORK

In recent years, there have been many proposed deep learning optimizers (Loshchilov & Hutter, 2019; Dozat, 2016; Zaheer et al., 2018; Phuong & Phong, 2019; Zhang & Mitliagkas, 2017; Jain et al., 2017; Zhuang et al., 2020; Zeiler, 2012; Shazeer & Stern, 2018; Tong et al., 2019; Ginsburg et al., 2019; Ma & Yarats, 2018; Lucas et al., 2018; Zhao et al., 2020; Chen et al., 2022a). In this extended related work section, we list the most recent deep learning optimizers that have influenced our proposed search space the most. AdaBelief (Zhuang et al., 2020) extends Adam by changing the update rule for the exponential moving average of the squared gradients by replacing the squared gradients term with the squared difference between the gradient and its momentum. AMSGrad (Phuong & Phong, 2019) extends Adam by updating the momentum term to be the maximum between the previous momentum term and the current time-step calculation. SADAM (Tong et al., 2019) extends Adam by incorporating a softplus activation around the square root of the exponential moving average of the squared gradients to calibrate the adaptive learning rate. Demon (Chen et al., 2022a) extends Adam by utilizing a decaying momentum schedule to $\beta$ for improved convergence. QHM (Ma & Yarats, 2018) replaces the gradient term in standard SGD with a linear combination between the gradient and the momentum term to introduce quasi-hyperbolic momentum. AggMo (Lucas et al., 2018) extends momentum SGD by taking the average between multiple momentum terms, each with their coefficient, to break oscillations through passive damping. We leverage these concepts by incorporating them in the creation of our new dual-joint search space.

## B EXTENDED DECAY FUNCTION

The proposed set of reduced unary operations for the decay function search space are listed in Table 7a, while the proposed set of decay schedules are listed in Table 7b.

| (a) Reduced set of unary operations. | (b) Proposed Decay Schedule Operands | |
|---|---|---|
| **Reduced Unary Operations** | **Decay Schedules** | |
| $x$ | $ld$ | $1 - t/T$ |
| $\text{sigmoid}(x)$ | $li$ | $t/T$ |
| $\frac{d}{dx}\text{sigmoid}(x)$ | $ldr$ | $1 - \mod(2t, T)/T$ |
| $\text{erf}(x)$ | $lir$ | $\mod(2t, T)/T$ |
| $\text{erfc}(x)$ | $cd$ | $0.5(1 + \cos(t\pi/T))$ |
| $\tanh(x)$ | $ci$ | $0.5(1 - \cos(t\pi/T))$ |
| $\arctan(x)$ | $cdr$ | $0.5(1 + \cos(\pi \mod(2t, T)/T))$ |
| $\text{bessel}_{i1e}(x)$ | $cir$ | $0.5(1 - \cos(\pi \mod(2t, T)/T))$ |
| $x^2$ | $ccd$ | $0.5(1 + \cos(2t\pi/T))$ |
| $\sqrt{x}$ | $cci$ | $0.5(1 - \cos(2t\pi/T))$ |
| $\text{softsign}(x)$ | $ed$ | $0.01^{(t/T)}$ |
| $\frac{d}{dx}\text{softsign}(x)$ | $ei$ | $1 - 0.01^{t/T}$ |
| $\frac{d}{dx}\tanh(x)$ | $dd$ | $\frac{(0.95*(1-t/T))}{(0.05+0.95*(1-t/T))}$ |
| | $di$ | $0.95 - \frac{0.95*(1-t/T)}{(0.05+0.95*(1-t/T))}$ |

Table 7: The proposed set of reduced unary operations for the decay functions, and the proposed set of decay schedule operands for the decay function search space. For the decay schedules, $t$ denotes the current timestep while $T$ denotes the maximum number of timesteps.

## C PARTICLE BASED GENETIC ALGORITHM

Pseudocode for the proposed particle based GA is given in Algorithm 1.

## D OPTIMIZER ELIMINATION PROTOCOL

Because we are unsure how well our optimizers trained on a small ConvNet would correlate to larger models, we devised an optimizer elimination protocol that progressively eliminated optimizers based

---

**Algorithm 1** Particle-Based Genetic Algorithm

---

1: **Input:** $n$ (Number of Particles), $k$ (Number of mutations), $t$ (Number of timesteps)
2: $initialParticles$ = OptimizerInitialization($100 \times n$)
3: $validationAccuracies$ = ConvNet($initialParticles$, base = 32)
4: $particles$ = BestParticles($initialParticles$, $validationAccuracies$, $n$)
5: **for** $particle$ in $particles$ **do**
6:   **for** $timestep$ in $t$ **do**
7:     $mutations = [\,]$
8:     **for** $j$ in $k$ **do**
9:       **repeat**
10:         $child$ = Mutation($particle$)
11:       **until** $child$.IntegrityCheck( )
12:       $mutations$.append($child$)
13:     **end for**
14:     $mutationValidationAccuracies$ = ConvNet($mutations$, base = 48)
15:     $particle$ = BestParticle($mutations$, $mutationValidationAccuracies$)
16:   **end for**
17: **end for**

---

upon their performance on increasingly larger ConvNets. For each run of our particle-based GA, we took the best 50 optimizers and trained them on a ConvNet with *base* 64 (0.375M parameters) for 16,000 steps. The best 24 were then taken and trained on a ConvNet with *base* 96 (0.839M parameters) for the same number of steps. The best 12 were then taken and trained with *base* 128 (1.487M parameters). Unlike the surrogate function used during evolution, the optimizer elimination protocol ran each optimizer three times for each ConvNet size to get an average of the validation accuracy on the CIFAR-10 dataset to use for comparison.

Finally, after performing optimizer elimination, the best six from each run were taken and trained on EfficientNetV2Small (20.3M parameters) using their proposed progressive RandAug regularization (Cubuk et al., 2020) for 64,000 steps. This was performed to ensure the best optimizers performed well on large-scale deep learning models. From the final 18 optimizers (best six from each of the three runs), the best six were hand selected based on their validation accuracy and uniqueness. An additional evolutionary run was performed for the fine tuning of each of the final six optimizers, described in Appendix E. The best 24 optimizers from those additional evolutionary runs were also evaluated on EfficientNetV2Small. The best 10 optimizers overall, from all optimizers evaluated on EfficientNetV2Small, were then reported as the final optimizers.

## E   ADDITIONAL EVOLUTIONARY RUNS

To ensure that the search space for each final optimizer had been completely exhausted, we took the final six optimizers from our proposed optimizer rejection protocol to be used as initial particles in our proposed GA. Each particle ran for $k = 12$, $t = 3$, and with $base = 64$. In addition, each child mutation was trained on the ConvNet three times to obtain a mean of the validation accuracy. This was performed to reduce the noise of training stochastic neural networks. The final best 24 optimizers from the six runs were evaluated on EfficientNEtV2Small. The best 10 optimizers from all optimizers evaluated on EfficientNetV2Small were then taken to be the final reported optimizers.

## F   OBSERVING THE FINAL DECAY FUNCTION AND LEARNING RATE SCHEDULES

The base learning rate schedule applied to all experiments was one cycle cosine decay with linear warmup. The learning rate is linearly warmed up to its peak, being held for a number of iterations, before being decayed using cosine decay back down to zero. All evolved optimizers learn to adapt this schedule through the use of distributed decay functions. For visualization, Figure 2a, Figure 2b, and Figure 2c plot the discovered learning rates over time. In each, the one cycle cosine decay with linear warmup base schedule is plotted in dashed lines for comparison. There appear to be two

families of learning rates (2a and 2b), along with two outsider learning rates (2c). Figure 2d plots the three internal decay functions used by the final optimizers, Opt4, Opt7/8, and Op10.

To assess how the three internal decay functions affect the weight update over time, in conjunction with the respective learned learning rate, eight time slices were plotted for the output distribution of the weight update. A one cycle learning rate totalling to 96,000 steps, with the first 6,400 being used for warm up and then being held for the next 12,800 before being cosine decayed, was used for obtaining the time slices. The eight time slices were recorded, at iterations 1K, 10K, 25K, 45K, 65K, 75K, 85K, and 93 or 95K. For example, for the internal decay function $t_i^3$ in Opt4, given an input of $x = e^{\hat{v}}$ ranging from 0 to 20, the output of $\text{clip}(2, t_i^3 x)$ times the learned learning rate of $t_i^1/t_i^2$ is plotted in Figure 3a. As one can see, this conjunction warms up to clipping 2 before being decayed over time, eventually clipping 2 to zero. Although it may seem that this learned internal decay function inside the clip operator with conjunction of the learned learning rate decreases the weight update value, it actually increases the weight update as it is used as the divisor of $(0.3g + 0.7\hat{v})$, where smaller values yield larger quotients. Unlike Figure 3a, Figures 3b, 3c, and 3d give the raw weight update values for optimizers 7, 8, and 10. Figures 3b, 3c, and 3d are extremely similar as they perform symmetric squashing functions on their respective inputs, greatly limiting the magnitude of the output. Scaling of the output is achieved by changing the maximum learning rate.

Optimizers Opt1, Opt2 and Opt3 can be seen as extensions of QHM, $QHM + \text{softsign}(x)$, where $x$ is dependent upon the optimizer. In an aid to assess how each $\text{softsign}(x)$ effects the weight update equation for their various variable inputs, Figure 4a, 4b, 4c gives the 3D plot for Opt1, Opt2, and Opt3. For example, the two primary inputs to $\text{softsign}(x)$ for Opt3 are the weights $w$ and the exponential moving average of the cubed gradients $0.01g^3 + 0.99\hat{\lambda}$. With these serving as the $x$ and $y$ axis, the output of $\text{softsign}(\frac{10^{-5}w}{\sqrt{1+(10^{-5}w-(0.01g^3+0.99\hat{\lambda}))^2}}))$ from Opt3 can be seen in Figure 4c. As one can see, the term only influences the weight update equation when the exponential moving average of the cubed gradients is centered around zero, where the sign and magnitude of $w$ directly control the sign and magnitude of the update.

Because the weight update equations for Opt4/5, Opt6, and Opt7 can be easily represented by two variable inputs, their 3D weight update equation with respect to their variable inputs is plotted in Figure 4d, 4e, and 4f.

## G  EXPERIMENTAL IMPLEMENTATIONS AND DETAILS

Six primary datasets were used in this work, CIFAR-10, CIFAR-100, Stanford-CARS196, Oxford-Flowers102, Caltech101, and Tiny. The meta-information for each of these datasets are available in Table 8.

| DATASET | TRAINING | EVALUATION | CLASSES | SIZE |
| --- | --- | --- | --- | --- |
| CIFAR10 | 50,000 | 10,000 | 10 | $32 \times 32$ |
| CIFAR100 | 50,000 | 10,000 | 100 | $32 \times 32$ |
| CARS196 | 8,144 | 8,041 | 196 | $224 \times 224$ |
| FLOWERS102 | 2,040 | 6,149 | 102 | $224 \times 224$ |
| CALTECH101 | 3,060 | 6,084 | 102 | $224 \times 224$ |
| TINYIMAGENET | 100,000 | 10,000 | 200 | $64 \times 64$ |

Table 8: The number of training and evaluation images, number of classes, and training image resolution for the datasets used in this work.

For each experiment, all optimizers and Adam-based learning rate schedules performed a learning rate test by selecting the best learning rate, from 10, 1, 0.1, 0.01, 0.001, 0.0001, and 0.00001, after a 1/3 of the training epochs (for Adam-based learning rate schedules, the learning rate test was performed for 2/3 of the training epochs). For CIFAR-10 and CIFAR-100, all optimizers were trained for 96,000 steps with a batch size of 64 on EffNetV2Small. Progressive RandAug regularization was used for image augmentation. For TinyImageNet, all optimizers were trained for 300,000 steps with a batch size of 256 on ResNet9 (6.5M parameters). For image augmentation, the images were resized using bicubic interpolation to 72x72, before a random 64x64 crop was applied, followed

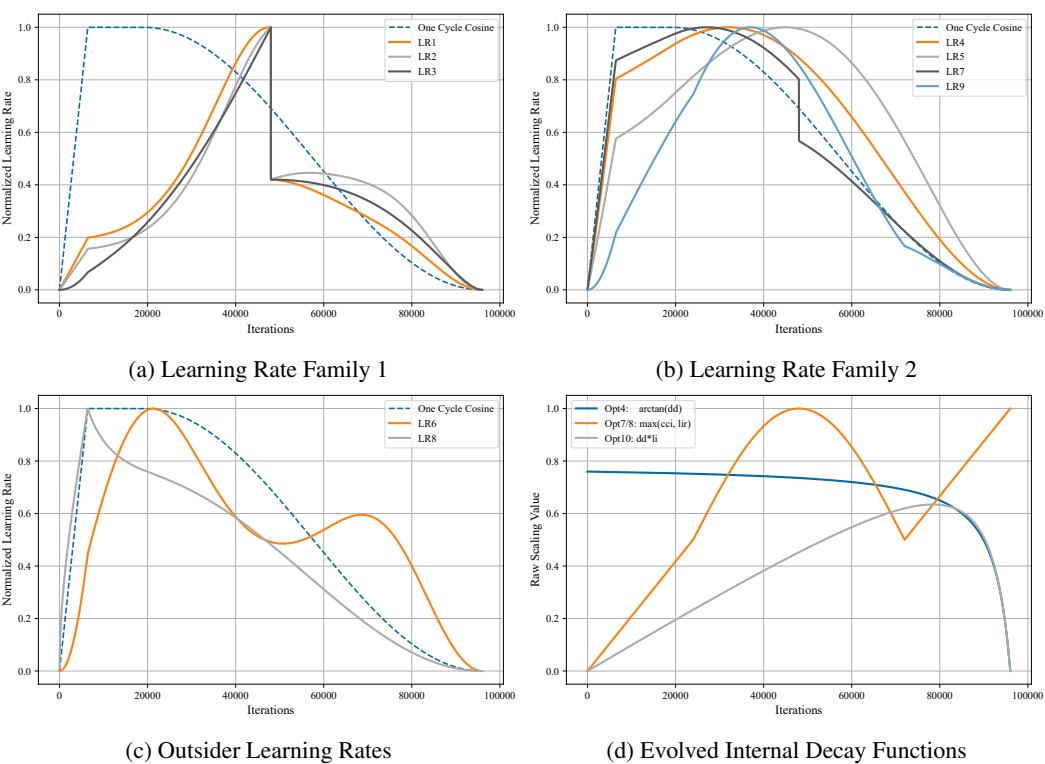

(a) Learning Rate Family 1

(b) Learning Rate Family 2

(c) Outsider Learning Rates

(d) Evolved Internal Decay Functions

Figure 2: (a)-(c) The final learning rate families found during evolution and from the supplementary evolution on Adam. Note, the learning rates are normalized between 0 and 1 to allow for comparison, the actual scaled differ quite drastically. (d) The three internal decay functions used in the evolved optimizers Opt4, Opt7/8, and Opt10.

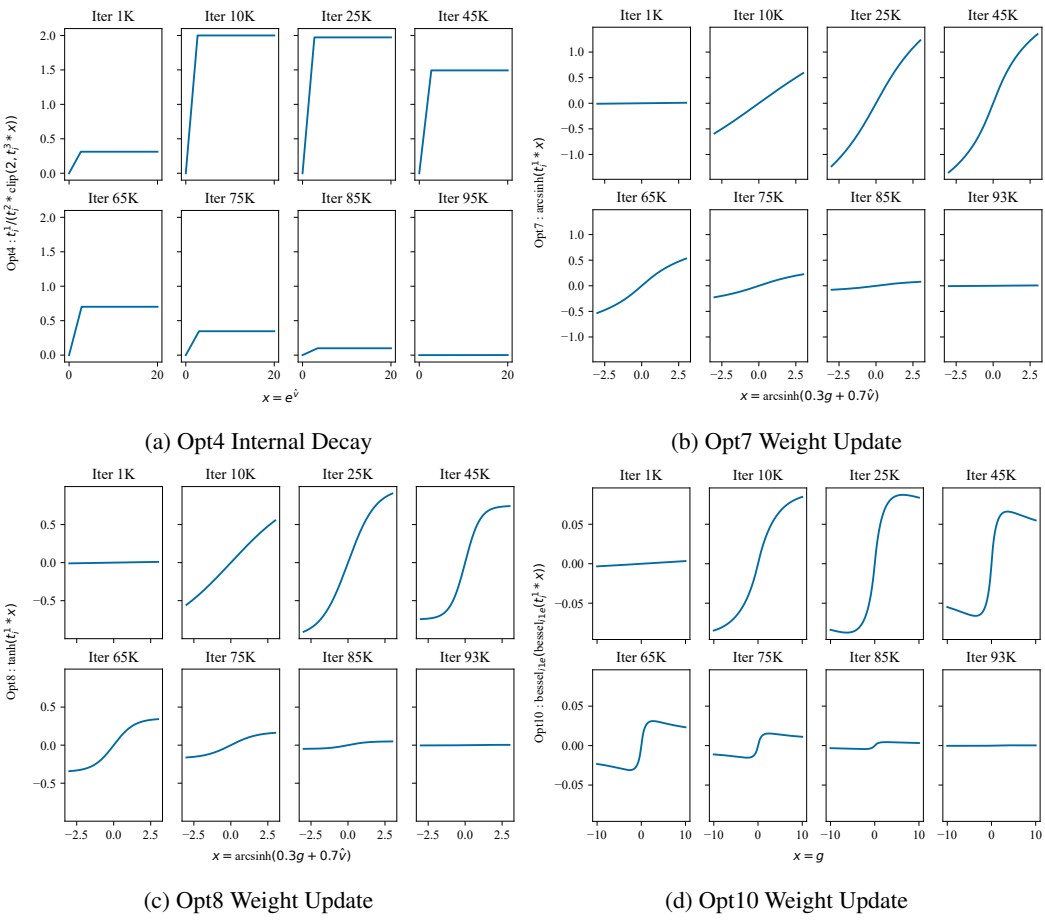

Figure 3: (a)-(d) Eight time slices (Iterations 1K, 10K, 25K, 45K, 65K, 75K, 85K, and 93 or 95K) of the effect of the discovered internal decay functions on the weight update for Opt4, Opt7, Opt8, and Opt10 over the course of a one cycle cosine decay learning rate schedule.

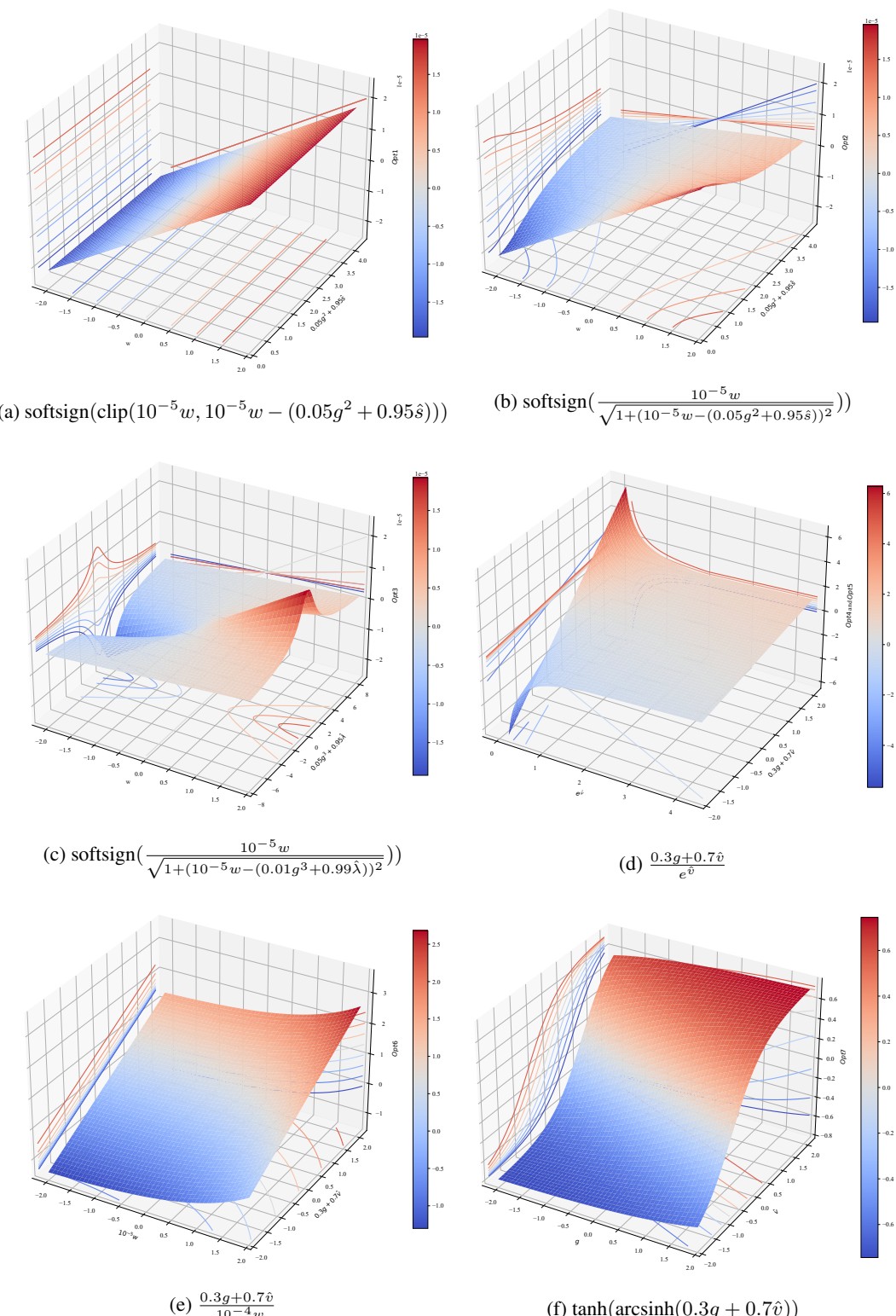

(a) softsign(clip($10^{-5}w, 10^{-5}w - (0.05g^2 + 0.95\hat{s})$)))

(b) softsign($\frac{10^{-5}w}{\sqrt{1+(10^{-5}w-(0.05g^2+0.95\hat{s}))^2}}$))

(c) softsign($\frac{10^{-5}w}{\sqrt{1+(10^{-5}w-(0.01g^3+0.99\hat{\lambda}))^2}}$))

(d) $\frac{0.3g+0.7\hat{v}}{e^{\hat{v}}}$

(e) $\frac{0.3g+0.7\hat{v}}{e^{10^{-4}w}}$

(f) tanh(arcsinh($0.3g + 0.7\hat{v}$))

Figure 4: (a-c) detail the output of softsign($x$) for the various $x$ of Opt1, Opt2, and Op3. (d-f) detail the weight update equation output for Opt4/5, Opt6, and Opt7 for their various inputs.

by random translation and horizontal flipping. For CIFAR-10, CIFAR-100, and TinyImageNet, the learning rate was linearly warmed up for 6,400 steps, being held for 12,800 steps at its maximum value, before being decayed with cosine annealing. For fine-tuning, the published ImageNet1K weights for EffNetV2Small were used as the initial weights before being fine-tuned for 10,000 steps on Flowers102, Cars196, and Caltech101 with a batch size of 64. For regularization, dropout of 0.5 was placed before the softmax layer, while RandAug regularization of magnitude 15 was used for image augmentation. For fine-tuning, all images were resized to 224x224. Due to the limited number of weight updates, the learning rate test was altered to fine-tune the learning rate. Specifically, the best found learning rate using the procedure described earlier was tested after being halved and multiplied by five. For example, a best found learning rate of 0.01 is tested at 0.05 and 0.005, where the final learning rate is chosen from this final set. All code was implemented in Tensorflow (Abadi et al., 2015), although the PTB experiments were performed in Pytorch (Paszke et al., 2019), and is freely available at the Github repository: *ANONYMOUS*.

## H   SUPPLEMENTARY PTB EXPERIMENTS

To assess generalization across deep learning domains, the final optimizers and learning rate schedules were transferred to language modeling on the PTB dataset. A three layer AWD-LSTM (Merity et al., 2017) with 1150 hidden units (24M parameters), softmax weight tying, variational dropout, embedding dropout, gradient norm clipping, truncated back propagation, and $L_2$ weight decay, was trained for 99,600 steps. The best validation perplexity for each optimizer is reported, where lower is better. The results are recorded in Table 9. From Table 9, three key points can be discussed. First, the best optimizer is arguably tied between Nesterov's momentum and $Opt10_1$, Opt10 without a decay function. $Opt10_1$ can be seen as an extension of Nesterov's momentum with weight gradient scaling. In addition, $Opt10_1$ greatly outperformed Opt10, possibly showcasing how the learned decay function is specifically designed for image recognition as Opt10 outperformed $Opt10_1$ on CIFAR. Second, all the discovered Adam variants outperform Adam for language modeling. Due to the clipping nature of the Adam variants, the absolute magnitude of the exponential moving average of the gradients is inherently reduced during the weight update, which prevents a moving average of exploding gradients from affecting the quality of the model, a common problem with LSTMs (Zhang et al., 2019). However, LR5, LR6, and LR7 were able to push standard Adam past all the Adam variants without the need of clipping. Third, despite Opt1, Opt2, and Opt3 being extensions of QHM, all outperformed QHM, indicating the importance the respective additional terms.

| OPTIMIZER | PTB |
|---|---|
| ADAM | 68.56±0.13 |
| RMSPROP | **67.47**±0.20 |
| SGD | 68.07±0.33 |
| NESTEROV | **65.33**±0.22 |
| POWERSIGN | 71.64±0.20 |
| ADDSIGN | 70.72±0.17 |
| QHM | 69.22±0.15 |
| OPT1 | **67.74**±0.10 |
| OPT2 | **67.62**±0.18 |
| OPT3 | **67.62**±0.15 |
| OPT4 | 69.81±0.21 |
| - OPT4$_1$ | **67.45**±0.12 |
| - OPT4$_2$ | 68.99±0.10 |
| OPT5 | 71.56±0.15 |
| OPT6 | 70.27±0.13 |
| - OPT6$_1$ | 69.25±0.22 |
| OPT7 | 70.47±0.14 |
| - OPT7$_1$ | 69.29±0.24 |
| OPT8 | 72.51±0.17 |
| - OPT8$_1$ | 69.11±0.19 |
| OPT9 | 68.73±0.19 |
| - OPT9$_1$ | 70.69±0.23 |
| OPT10 | 71.99±0.15 |
| - OPT10$_1$ | **65.34**±0.15 |
| LR1 | 73.19±0.26 |
| LR2 | 72.14±0.31 |
| LR3 | 71.82±0.33 |
| LR4 | 73.31±0.32 |
| LR5 | 67.87±0.07 |
| LR6 | **67.84**±0.11 |
| LR7 | 68.01±0.15 |
| LR8 | 73.20±0.12 |
| LR9 | 72.57±0.20 |
| A1 | 68.26±0.20 |
| A2 | 68.22±0.12 |
| A3 | 68.28±0.12 |
| A4 | 68.22±0.14 |
| A5 | 68.41±0.21 |

Table 9: Results for all optimizers and learning rate schedules for PTB. The mean and standard deviation of the best validation perplexity is reported (lower is better) across 5 independent runs are reported. Red indicates Top 3 performance, while bold indicates Top 8 performance.

