# OpenReview forum: "Neural Optimizer Equation, Decay Function, and Learning Rate Schedule Joint Evolution"
_ICLR.cc/2024/Conference — Submitted to ICLR 2024_

### Official Review · Reviewer_8NVV · 2023-10-29

**Soundness:** 3 good
**Presentation:** 2 fair
**Contribution:** 2 fair
**Rating:** 3
**Confidence:** 4

**Summary:**

This paper focuses on the automated discovery of optimization algorithms. In particular, it proposes an enlarged optimizer search space constructed using tree-based grammar using building blocks of both update equations as well as decay functions (functions of the optimization step number). To search over this space, the authors employ a mutation-only genetic algorithm that evolves optimizers by mutating and selecting the best optimizer according to its generalization performance on the Cifar-10 dataset. The authors also propose a sanity check to eliminate degenerate optimizers by first checking its performance when training on a quadratic task to save compute. The discovered optimizers are shown to outperform or match existing hand-designed optimizers on training/fine-tuning EffNetV2Small on multiple vision datasets.

**Strengths:**

- Automated discovery of optimization algorithms is an important topic.
- The paper provides a thorough documentation of its algorithm procedure.
- The proposed sanity check of first evaluating the optimizer on a quadratic task is reasonable and can potentially save compute.

**Weaknesses:**

1. __Significance in the Contribution__. To my understanding, the two types of contributions the authors can make in this paper are: __1)__ proposing a new set of algorithm procedures (including new search space, search algorithm, integrity check) to find better optimizers; __2)__ the actual optimizers found and any insights obtained from analyzing them.

    In terms of __1)__, despite the authors doing a generally good job of documenting their procedures for reproducibility, the actual procedures are very specialized to the ConvNet model architecture (with multiple rounds of different types of heuristics to filter the search candidates) and might still be difficult to replicate and also generalize to a new task, architecture combination.

    In terms of __2)__, it’s not clear to me how much better the discovered optimizers are compared to existing hand-designed optimizers like Adam. All of the results in Table 6 are evaluated using a single model architecture EffNetV2Small after performing some final optimizer candidate selection directly over this model architecture. Therefore, it’s unclear to what degree these optimizers can general to other models. Besides, focusing on EffNetV2Small, over the non-cifar tasks (tasks not used in the optimizer selection), there always exist hand-designed optimizers that are close if not better than the discovered ones. Finally, in terms of interpreting the analytical form of the discovered optimizers, the paragraph at the end of page 8 and beginning of page 9 are more of surface-level descriptions of the experimental results rather than a distilled summary and insights, making it difficult to parse the key message the authors wish to convey.

2. **Evaluating the importance of enlarging the search space**. As one of the claimed contributions of the paper is an enlarged optimizer search with more operands, operators, and decay functions, it is expected that the authors should perform an ablation study to understand the benefit of the introduced enlarged search space. The bare-minimum ablation baseline is to train on the datasets in Table 6 using several of the optimizers discovered by Bello et al (2017) and compare whether the performances of the optimizers discovered in this paper are better than those discovered in a smaller search space.

3. **Incomplete results**. Part of the results in Table 6 are unfinished as the authors claim to finish before the rebuttal period.

**Questions:**

- In Bello et al (2017), in addition to tables comparing optimizers’ performance, learning curves of optimizers’ generalization performance progression are also shown. For some of the cases, it helps reader to see that Bello et al's discovered optimizers can converge faster than hand-designed methods for cases where the final performances might be similar. I suggest the authors also consider showing such graphs to provide additional information of the discovered optimizers.
- The authors should formally define what a decay function is. I personally found it hard to understand the terminology until I read its definition in Bello et al (2017).
- The authors mention that the mutation operator in the paper selects the best mutated child as the next position for the particle. It’s not clear to me whether this best child is compared against its parent. Can the authors clarify if this is done? (If so, then the selection should guarantee monotonic performance improvement.)
- On page 4, the authors use the term “non-distributive function”. However, I wonder if the authors mean to say non-homogeneous function.

---

> ### Author Response · Authors · 2023-11-13
>
> Hello Reviewer,
>
> Thank you for taking the time to detail your thoughts, concerns, and grading. We appreciate your criticism in making our work more professional, robust, and clear. Please see our official comment (**Updated Unfinished Results**) on the subject of unfinished results.
>
> > 1. Significance in the Contribution.
>
> > the actual procedures are very specialized to the ConvNet model architecture (with multiple rounds of different types of heuristics to filter the search candidates) and might still be difficult to replicate and also generalize to a new task, architecture combination.
>
> It is common in AutoML tasks applied to neural networks to use a proxy function as training and evaluating thousands of large scale models is infeasible. Small ConvNets have been historically used as such a proxy function. Bello et al (2017) used a ConvNet when searching for their optimizers. [1] used a convnet when evolving activation functions. [2]  used a convnet when evolving loss functions. It is typical to take the best handful from the convnets and then train them large scale, as was performed by Bello et al (2017). The only difference between what we did and Bello et al (2017), was use an elimination protocol that selected the best handful by progressively upscaling the ConvNet. This was performed as our implementation of the optimizer graph was extremely memory intensive, as we had to keep track of atleast 4x the size of the model in memory ($\hat v, \hat s, \hat \lambda$ and momentum) (not counting internal state operands), which did not fit into memory when using EfficientNetV2Small given our computational resources. All final optimizers had to be hard-coded into memory efficient manners to allow for training upon EffNetV2Small. Therefore, we had to devise a way to select the most capable from the discovered optimizers other than hard-coding 150 optimizers.
>
> > "it’s not clear to me how much better the discovered optimizers are compared to existing hand-designed optimizers like Adam."
>
> Please see our official comment (**Updated Unfinished Results**) as we have filled in the missing experiments. We would like to note that when training from scratch on CIFAR-10, CIFAR-100, and TinyImageNet, Opt6 greatly outperformed Adam. When fine-tuning, Opt3 greatly outperformed Adam. In addition, A1 and A5 outperformed Adam across all datasets (except CIFAR-10).
>
> >  Finally, in terms of interpreting the analytical form of the discovered optimizers, the paragraph at the end of page 8 and beginning of page 9 are more of surface-level descriptions of the experimental results rather than a distilled summary and insights, making it difficult to parse the key message the authors wish to convey.
>
> Please see our official comment (Convergence, Sensitivity, and Theoretical Bounds for Found Deep Learning Optimizers) on the subject. There is not enough room in 9 pages to detail every insight of the final discovered optimizers. Please see Appendix G for in-depth discussion on the topic.
>
> > "All of the results in Table 6 are evaluated using a single model architecture EffNetV2Small after performing some final optimizer candidate selection directly over this model architecture. Therefore, it’s unclear to what degree these optimizers can general to other models."
>
> This is semi-incorrect. Although it is true that the final optimizers were selected based upon their performance on EffNetV2Small, Table 6 does contain results on a different architecture: A ResNet9 (6M parameters) for TinyImageNet. In addition, Appendix H gives a supplementary experiment on training an LSTM (24M parameters) on the PTB dataset.
>
> > 2. Evaluating the importance of enlarging the search space.
>
> Please see our official comment (**Updated Unfinished Results**) on the subject of unfinished results as we have included the best two discovered optimizers from Bello et al (2017) in our work.
>
> > it is expected that the authors should perform an ablation study to understand the benefit of the introduced enlarged search space
>
> Unfortunately, we did not have the computational resources to run ablation studies on different combinations of the search space.
>
> > 3. Incomplete results.
>
> Please see our official comment (**Updated Unfinished Results**) on the subject of unfinished results.
>
> [1] Garrett Bingham, William Macke, and Risto Miikkulainen. Evolutionary optimization of deep learning activation functions. In Proceedings of the 2020 Genetic and Evolutionary Computation Conference, GECCO ’20, page 289–296, New York, NY, USA, 2020. Association for Computing Machinery. ISBN 9781450371285. doi: 10.1145/3377930.3389841. URL https://doi.org/10.1145/3377930.3389841.
>
> [2] Santiago Gonzalez and Risto Miikkulainen. Improved training speed, accuracy, and data utilization through loss function optimization. In 2020 IEEE Congress on Evolutionary Computation (CEC), pages 1–8, 2020. doi: 10.1109/CEC48606.2020.9185777.

---

> > ### Author Response · Authors · 2023-11-13
> >
> > > " I suggest the authors also consider showing such graphs to provide additional information of the discovered optimizers."
> >
> > Unfortunately, we did not keep the training plots, but only their best test results. As a result, we do not have the computational resources to run every optimizer on each dataset again to provide this material by the end of the deadline.
> >
> > > "The authors mention that the mutation operator in the paper selects the best mutated child as the next position for the particle. It’s not clear to me whether this best child is compared against its parent. Can the authors clarify if this is done? (If so, then the selection should guarantee monotonic performance improvement."
> >
> > The best child is not compared against its parent. This is correct. We did not use our genetic algorithm in the classic sense, where the best stays within the population (or as the main particle in our scenario) and after convergence the best are chosen as the final optimizers. Real et al (2018) showed that incorporating aging within their algorithm performed the best. Then after evolution, the best optimizers found over time are taken as the best. We used our proposed genetic algorithm more as an exploratory mechanism, from which our proposed optimizer elimination protocol exploited the best optimizers found after evolution.
> >
> > > "On page 4, the authors use the term “non-distributive function”. However, I wonder if the authors mean to say non-homogeneous function."
> >
> > We refer to a "non-distributed function" as any function such that $f(c*x) \ne c * f(x) $, an example would be
> >
> > $ \tan( c * x ) \ne c*\tan(x) \forall c \ge 0$.

---

### Official Review · Reviewer_Mvi5 · 2023-10-30

**Soundness:** 2 fair
**Presentation:** 1 poor
**Contribution:** 1 poor
**Rating:** 3
**Confidence:** 4

**Summary:**

This paper proposed to evolve an optimizer for neural networks. The researched problem is necessary, while the technical contributions are limited. In addition, the experiments should be conducted on SOTA neural network models for verification.

**Strengths:**

Designig suitable optimizers for specific problems is necessary, while this work is just for this aspect.

**Weaknesses:**

There are many types of genetic algorithms in the literature, while the proposed genetic algorithm in this paper is made based on the claim that the genetic algorithm used in Real et al.'s work is not suitable for the work in this paper. In fact, many very similar evolutionary algorithms can achieve the same goals (mutation only, aging, parallelism) as the designed genetic algorithm in this paper.

This paper looks like a project implementation, instead of a research paper. To achieve the goal claimed in this paper, the authors prepare a lot of different components for the project's implementation.

The convenient way to verify if the proposed method works for the current fact is to search an optimizer on some SOTA neural networks, and then to check if the performance of the compared SOTA can be improved.

The format of references should be updated, the current form of submission is hard to read.

**Questions:**

See above

---

> ### Author Response · Authors · 2023-11-13
>
> Hello Reviewer,
>
> Thank you for taking the time to detail your thoughts, concerns, and grading. We appreciate your criticism in making our work more professional, robust, and clear. Please see our official comment (**Updated Unfinished Results**) on the subject of unfinished results.
>
> > 1. There are many types of genetic algorithms in the literature, while the proposed genetic algorithm in this paper is made based on the claim that the genetic algorithm used in Real et al.'s work is not suitable for the work in this paper. In fact, many very similar evolutionary algorithms can achieve the same goals (mutation only, aging, parallelism) as the designed genetic algorithm in this paper.
>
> We are aware that the literature is filled with a plethora of different genetic algorithms. We have no problem using the genetic algorithm from Real et al.'s work, except that it yielded repeat optimizers most of the time in our preliminary experiments, due to the reasons discussed in the paper. Therefore, we had to use a different algorithm to search the proposed search space. Due to the success of Real et al.'s work, we modeled our algorithm after the main attributes of their algorithm. We do not see how this is a weakness as the main contribution of our work was not the genetic algorithm but the deep learning optimizer search space and discovered optimizers.
>
> > 2. "This paper looks like a project implementation, instead of a research paper."
>
>  The research question behind our paper can be formulated as follows: "Can we use AutoML to discover deep learning optimizers that can achieve SOTA results compared to baseline human designed deep learning optimizers in computer vision?" This is the backbone question for many AutoML papers, especially neural architecture search papers: "Can we use AutoML, and a cleverly designed search space and search algorithm, to find an architecture that outperforms human engineered architectures?".
>
>
> > 3. "The convenient way to verify if the proposed method works for the current fact is to search an optimizer on some SOTA neural networks, and then to check if the performance of the compared SOTA can be improved."
>
> This is correct. In fact, that is exactly what we have done. After searching for optimizers, the best were transferred to EfficientNetV2Small, a SOTA computer vision CNN, across multiple image classification datasets: CIFAR-10, CIFAR-100, Flowers, Cars, Caltech, and Tiny. We found multiple optimizers that consistently outperformed the baseline SOTA deep learning optimizers. Please see our discussion in Section 6 on the results.
>
> > 4. The format of references should be updated, the current form of submission is hard to read.
>
> Thank you reviewer for pointing that out, we have updated our citations in our new revision.

---

### Official Review · Reviewer_XE1e · 2023-10-31

**Soundness:** 2 fair
**Presentation:** 2 fair
**Contribution:** 2 fair
**Rating:** 3
**Confidence:** 4

**Summary:**

The paper claims simultaneous optimization of the weight update equation, decay functions, and adaptation of the learning rate schedule. Certain operands are tested on standard data sets and a particle GA method for simultaneous optimization is proposed.

**Strengths:**

The simultaneous optimization approach is interesting.

**Weaknesses:**

The approach lacks clarity.
It doesn't provide an theoretical guarantees of convergence neither does it talk about algorithmic complexity. The paper lacks theoretical soundness.

**Questions:**

For example, how does the learning rate adapt? With the Particle-GA?
Not sure why the LRs are bumpy

---

> ### Author Response · Authors · 2023-11-13
>
> Hello Reviewer,
>
> Thank you for taking the time to detail your thoughts, concerns, and grading. We appreciate your criticism in making our work more professional, robust, and clear. Please see our official comment (**Updated Unfinished Results**) on the subject of unfinished results.
>
> > 1. "It doesn't provide an theoretical guarantees of convergence neither does it talk about algorithmic complexity"
>
> Please see our official comment (**Convergence, Sensitivity, and Theoretical Bounds for Found Deep Learning Optimizers**) on the subject.
>
> > 2. " how does the learning rate adapt?"
>
> The learning rates were tested for generalization by training them across multiple image classification datasets. In addition, there were also tested in language modeling on the PTB dataset using an LSTM. These results were discussed in Appendix H.
>
> > 3. "With the Particle-GA? "
>
> Our goal was not propose a particle based GA that could generalize to other scenarios, but to utilize the proposed approach specifically for our task of discovering deep learning optimizers.
>
> > 4. "Not sure why the LRs are bumpy"
>
> The learning rates are bumpy because that is what the particle based genetic algorithm found successful. This can be demonstrated in Table 6 for CIFAR-10 where four of the learning rates achieve Top~8. The "bump[iness]" of the discovered learning rates, in conjunction with its success on CIFAR-10 indicates and challenges the assumption that smooth learning rates are most optimal.

---

### Official Review · Reviewer_hhkB · 2023-11-01

**Soundness:** 2 fair
**Presentation:** 2 fair
**Contribution:** 2 fair
**Rating:** 3
**Confidence:** 4

**Summary:**

The authors investigate hyperparameter tuning in machine learning models. I think while the topic is a well-explored area, the continuous evolution of models and the increasing complexity of tasks mean that there's space for innovative approaches. The paper proposes the use of Neural Optimizer Search (NOS) in this task. The experiments ranked ten final optimizers, categorized into three main families. I do like the transferability experiments, where the algorithms are tested across various image classification tasks. The performance varies between models trained from scratch and fine-tuning scenarios, with some optimizers showing particularly strong results in one or the other. But as in many other papers in this context, I think the why the algorithms behave like that is not entirely clear.

**Strengths:**

I do think the authors presented an interesting approach by expanding the search space for optimizer algorithms and introducing a novel particle-based GA method.
The authors also designed a comprehensive and extensive set of experiments, but I think increased the ablation study to help evaluate the contributions of individual components in the proposed investigation.
The paper is well-organized, I liked the approach of breaking down the problem, its causes, the solution, and the experimental validation in a logical sequence.

**Weaknesses:**

Besides a large number of experiments, I think the main experiments are centered on or in variations of specific datasets like TinyImageNet and CIFAR-10.

One relevant weakness of the paper is that a relevant quantity of experiments were not completed at the time of the paper's writing, as denoted by asterisks in Table 6. The lack of results leaves a gap in the comprehensive evaluation of the results and makes it hard to evaluate the global performance of the proposed approach.

The authors focused experiments on image classification and language modeling tasks but argued that the results could be extended to other tasks.

I was waiting for a more in-depth analysis of the results, considering the whys, the sensitivity analysis, tendencies, etc. The paper shows the phenomena exist but does not explain them in detail. I know this is common in papers in this field, but we need to improve that. It has an interesting empirical contribution but lacks theoretical support for why certain optimizers perform better than others.

The fine-tuning results appear to be inconsistent with the from-scratch training results.

I do believe the authors benefit from a discussion on scenarios where the proposed method might not work well. This gives readers a more balanced view and sets expectations correctly. I suggest the authors explore more of that.

The paper identifies families of optimizers, but it doesn't explore the unique characteristics of each family and how they contribute to the model's performance.

**Questions:**

The paper focuses on EfficientNet and ResNet, what about the performance in different architectures such as transformers or RNNs?

Given the stochastic nature of genetic algorithms, how consistent are the results across different runs of the optimizer discovery process?

Can the authors provide details on the computational resources required for the optimizer discovery process?

How do the optimizers perform under adverse training conditions, such as noisy gradients, sparse gradients, or data with high-class imbalance?

I noted seven of the final ten optimizers contain decay functions, what is the specific contribution of these decay functions to the overall performance?

---

> ### Author Response · Authors · 2023-11-13
>
> Hello Reviewer,
>
> Thank you for taking the time to detail your thoughts, concerns, and grading. We appreciate your criticism in making our work more professional, robust, and clear. Please see our official comment (**Updated Unfinished Results**) on the subject of unfinished results.
>
> > 1. "I was waiting for a more in-depth analysis of the results, considering the whys, the sensitivity analysis, tendencies, etc."
>
> Please see our official comment (**Convergence, Sensitivity, and Theoretical Bounds for Found Deep Learning Optimizers**)  on the subject.
>
> > 2. "The fine-tuning results appear to be inconsistent with the from-scratch training results."
>
> Yes, this observation does occur as the performance of the optimizers flip when training from scratch and fine-tuning. This also occurs for the baseline deep learning optimizers, such as SGD and QHM. In addition, top performing optimizers in each scenario do not flop in the other. For example, Op6, the best performer when training from scratch, performs comparably to Adam on Flowers and Caltech; and, Opt3, arguably the best performer when fine-tuning, outperformed Adam on CIFAR100 and Tiny. Therefore, we do not see this as a weakness, but an exposition of the strengths and weaknesses for different optimizers.
>
> > 3. "The paper identifies families of optimizers, but it doesn't explore the unique characteristics of each family and how they contribute to the model's performance."
>
> Please see our official comment (**Convergence, Sensitivity, and Theoretical Bounds for Found Deep Learning Optimizers**)  on the subject. In addition, Appendix F discusses the effects of the final decay functions, learning rate schedules, and weight update equations.
>
> > 4. "The paper focuses on EfficientNet and ResNet, what about the performance in different architectures such as transformers or RNNs?"
>
> Unfortunately, we did not have enough computational resources to test each optimizer across different deep learning domains. However, our submission does include results for training an LSTM on the PTB dataset, as recorded in Table 9.
>
> > 5. "Given the stochastic nature of genetic algorithms, how consistent are the results across different runs of the optimizer discovery process?"
>
> Our proposed algorithm was more exploratory than exploitive. We did not use our genetic algorithm in the classic sense, where the algorithm is ran and the best/final individuals from the last generation are taken as the best. We used our proposed genetic algorithm more as an exploratory mechanism, from which our proposed optimizer elimination protocol exploited the best.
>
> > 6. "Can the authors provide details on the computational resources required for the optimizer discovery process?"
>
> As quoted from our paper: "we only had access to one NVIDIA 4090 GPU during evolution. Each run took approximately six days to complete".
>
> > 7. "How do the optimizers perform under adverse training conditions, such as noisy gradients, sparse gradients, or data with high-class imbalance?"
>
> Please see our official comment (**Convergence, Sensitivity, and Theoretical Bounds for Found Deep Learning Optimizers**)  on the subject.
>
> > 8. "I noted seven of the final ten optimizers contain decay functions, what is the specific contribution of these decay functions to the overall performance?"
>
> Section 5 discussed that each optimizer which learned a decay function was trained both with and without it to assess its contribution. These are included with the results by the denotation of $_1$ to the end of the optimizer. For example, Opt6 trained without all of its decay functions is denoted by Opt6$_1$. Unfortunately, we did not have the time, nor space, to train every combination of optimizer and decay function, therefore we trained each optimizer both with and without all of its discovered decay functions.

---

### Author Response · Authors · 2023-11-13
**Updated Unfinished Results**

Hello Reviewers,

We would like to notify the reviewers that we have updated our submission with the unfinished results for Table 6. Specifically, we have filled in the * entries. In addition, we noticed an error in our code for testing Opt6$_1$, Opt7$_1$, Op8t$_1$, Opt9$_1$, and Opt10$_1$ for Flowers and Caltech. We have reran those experiments and updated their results. In addition, at the request and suggestion of some reviewers, we have taken the best two found deep learning optimizers from Bello et al, (2017); namely, PowerSign and AddSign, both using linear decay, to use as baseline optimizers. Lastly, because our final found optimizers heavily used $(0.3g + 0.7 \hat v)$, which by definition is the quasi-hyperbolic momentum optimizer (QHM), Ma & Yarats, 2018; we have also included it in our results. Besides grammatical fixes, Section 5 (Transferability Experiments) was split into Section 5 (Transferability Experiments) and 6 (Results and Discussion) to better flow with the paper. In addition, we have moved section 3.6 Optimizer Elimination Protocol to our Appendix in order to make room for more discussion in Section 6 on the final discovered optimizers as requested and suggested by some reviewers. Lastly, at the request and suggestion by some reviewers, we have updated our appendix to include a deeper discussion over the effect of different components for each discovered optimizer.

We urge reviewers to reread Section 5 and 6, along with Appendix E for further discussion.

In addition, it seems that some reviewers were unaware that we also performed a supplementary experiment by training each of the optimizers for language modeling using an LSTM (24M parameters) on the PTB dataset. This was included in Appendix H. Unfortunately, we did not have room to include it within the main body of the text, therefore it was pushed to the Appendix, and was mentioned in Section 5. We suggest the reviewers view the results discussed in Appendix H.

---

### Author Response · Authors · 2023-11-13
**Convergence, Sensitivity, and Theoretical Bounds for Found Deep Learning Optimizers**

Hello Reviewers:

Multiple reviewers have discussed a major weakness of the paper to be a lack of ``` in-depth analysis of the results, considering the whys, the sensitivity analysis, tendencies, etc.```,  ```It doesn't provide an theoretical guarantees of convergence neither does it talk about algorithmic complexity.```, and finally no discussion on ```interpreting the analytical form of the discovered optimizers```.

We would like to answer these concerns as an official comment instead of copy-pasting to each rebuttal. Entire papers have been published with the sole devotion to studying the effects, convergence, and quirks of a single deep learning optimizer. The literature is filled with them. In this paper, we found and discussed over 15 unique deep learning optimizers. We believe that it is not within the scope of this work to thoroughly discuss the effect, convergence, and sensitivity that each component contributes for each found optimizer in different scenarios. In itself, that could contain multiple papers.

Following the work of Bello et al (2017), our objective was to introduce a new search space to allow for complex and unique deep learning optimizers that showed empirical success compared to baseline optimizers. Using the most up-to-date research on deep learning optimizers, we have constructed a search space specifically designed for deep learning optimizers. After applying our proposed particle-based genetic algorithm, we found multiple novel optimizers, learning rates schedules, and Adam variants that outperformed the baseline deep learning optimizers across various image classification datasets.

---

### Meta-Review · Area_Chair_WLa1 · 2023-12-10

**Metareview:**

This paper proposes to use automated search to discover novel and high-performing optimization algorithms for deep learning models. The proposed search space and algorithm allow optimizing multiple components jointly. Empirical results demonstrate improved accuracy over human-designed optimizers on visual tasks.

There raised concerns about lack of theoretical analysis, limited architectures tested, and inconsistent fine-tuning results, and thus  the contribution was incremental and results were incomplete at time of submission. In response, the authors provided additional experiments and analysis to address the reviewers' concerns around theoretical grounding, sensitivity analysis, generalizability, and comparison to prior work. They acknowledged limitations around compute resources for additional ablation studies.

Overall, the reviewers raised important points about significance of contribution and thoroughness of evaluation. The authors made a good faith effort to address the feedback. However, the core weaknesses around novelty and rigor do not seem fully resolved.

**Justification For Why Not Higher Score:**

The contribution of this paper is not recognized.

**Justification For Why Not Lower Score:**

N/A

---

### Decision · Program_Chairs · 2024-01-16

Reject